# A Meta-Analysis on the Effectiveness of Gratitude Promotion Programs for South Korean College Students

**DOI:** 10.3390/bs14030240

**Published:** 2024-03-15

**Authors:** Namki Lee, Yucheon Kim

**Affiliations:** Department of Counseling and Coaching, Dongguk University, 30, Pildong-ro 1 Gil, Jung-gu, Seoul 04620, Republic of Korea; nearman7@dgu.ac.kr

**Keywords:** college students, gratitude, meta-analysis, cognitive effects, emotional effects

## Abstract

This study delves into the importance and consequences of gratitude promotion programs for South Korean college students. It uses meta-analyses to evaluate the effects of such programs on this demographic, shedding light on their significance and implications. To that end, we selected 11 papers in which 683 persons participated as study subjects, analyzing them using the PRISMA methodology. We observed an overall effect size of 0.6671, categorized as above medium. The effect sizes of the sub-areas were cognitive (d = 1.37), emotional (d = 0.60), and behavioral (d = 0.77), with the cognitive area exhibiting the largest effect size. When categorized by program type, the effect size (d = 0.85) for the program involving gratitude practice and gratitude recording surpassed the effect size (d = 0.77) of the programs where participants solely engaged in gratitude recording. According to program execution durations, the largest effect size (d = 1.61) appeared in the group that executed the program for the longest period of at least 16 weeks. This study highlights the areas where gratitude promotion programs for college students prove to be most effective. These findings offer valuable insights for tailoring and developing gratitude promotion programs in the future.

## 1. Introduction

Positive psychology, which studies human happiness, indicates that it is important to discover the meaning of life by developing positive emotions and strengths rather than eliminating the negative emotions in our inner lives [1]. Positive psychology divides positive human characteristics into 6 virtues and 24 strengths. Gratitude belongs to the transcendence category among the six virtues, which are wisdom, humanity, courage, temperance, justice, and transcendence [2]. In positive psychology, gratitude occupies a major part, and some scholars [3] consider positive psychology to be an attempt to give humans a grateful perspective on their potential, motivation, and ability.

Gratitude is one of the common themes in explaining religious life [4]. Scholars have emphasized it as a core virtue that individuals must possess in major religions [5]. Throughout time, people have passed down gratitude as the wisdom of life, considering it a desirable aspect of the human character and in life [6]. People with a high gratitude disposition are highly satisfied with their lives [7,8], frequently behaving favorably and pro-socially [9]. In addition, gratitude increases positive emotions and reduces negative emotions [10].

Meanwhile, time at university is important for establishing one’s ego identity [11]. However, early adulthood is the most difficult time in life, when individuals must accomplish important tasks for independence while receiving little attention and help from the people around them [12]. For such reasons, college students have a tendency towards depression that is twice as high as that of office workers of similar ages [13]. However, college students have limited opportunities to deal with their future through proper programs [14]. Amid the challenges college students face, research has shown that cultivating gratitude positively influences their overall college experience, including academic achievement and satisfaction [15]. Moreover, it fosters social support within their peer relationships and school lives, contributing to developing strong bonds with the school [16]. In addition, gratitude enables college students to overcome psychological difficulties and feel happy [17]. Since gratitude can have many positive effects on college students, we became interested in the effectiveness of gratitude promotion programs on various psychological variables.

The present study employs meta-analysis to examine and synthesize the effectiveness of each included study. Meta-analysis is a method employed to integrate scattered study results, providing comprehensive answers to the researcher’s questions and deriving statistical analysis results from the combined data [18]. It synthesizes the results of studies and analyzes their effectiveness to derive objective results [19]. Therefore, a meta-analysis can present comprehensive results without focusing on only a few study results, and it can find the causes when studies produce different results [20].

Thus far, two meta-analysis studies relate to gratitude in South Korea. Moon [21] conducted a meta-analysis of gratitude group counseling programs but did not limit the study subjects, only analyzing programs related to counseling. In addition, Shin and Kim [22] conducted a meta-analysis on gratitude-related variables, studying the effectiveness of related variables. However, their study did not limit the subjects, and the researchers did not analyze gratitude promotion programs. Gratitude related to happiness is necessary for all ages, and even more so for college students with many psychological difficulties. Therefore, a meta-study that can comprehensively examine individual gratitude-related studies conducted with college students is necessary. Given the absence of meta-analyses concerning gratitude promotion programs for South Korean college students, we undertook a meta-analysis to assess the effectiveness of such programs in enhancing the happiness and quality of life among college students. Through this study, we thoroughly examined the significance of college students’ gratitude promotion programs, enabling us to propose more effective methods for designing and implementing such programs. Thus, this study’s research questions are as follows:

RQ1. What is the overall effect size of the gratitude promotion program for college students?

RQ2. What are the effect sizes of variables on cognition, emotion, and behavior, which are sub-areas of the effectiveness of the gratitude promotion program?

## 2. Study Method

This study conducts a meta-analysis to assess the effectiveness of gratitude promotion programs for South Korean college students. It achieves this by analyzing and synthesizing individual studies related to these programs.

### 2.1. Study Inclusion and Exclusion Criteria

We conducted this study by applying the PICOS framework (population, intervention, comparison, outcomes, study design) [23] and refining the research topic into concrete research questions. The population (P) is South Korean college students, intervention (I) is gratitude promotion programs, and we conducted a comparison (C) with an active control group that contrasted with the experimental group. We obtained outcomes (O) by analyzing the overall effectiveness of the dependent variables that affected college students and the effectiveness of cognition, emotion, and behavior-related variables. We set the study design (S) as an experimental group-control group pretest-posttest.

This study exclusively considered papers listed in the Korean Journal Citation Index (KCI) to enhance the credibility of the meta-analysis while excluding those with insufficient data, qualitative studies, and papers not focused on college students.

### 2.2. Search Strategy

In September 2023, we searched all theses and papers published in academic journals using the search engines and databases of the Research Information Sharing Service (RISS), Korean Studies Information (KISS), and the National Assembly Library of the Republic of Korea (NAL). We used the keywords “gratitude”, “thankfulness”, and “appreciation” to search abstracts in September 2023 to collect primary data, then using the same keywords to collect secondary supplementary and additional data in October 2023. We registered our meta-analysis protocol in the International Prospective Register of Systematic Reviews (registration number CRD42024497664, 15 January 2024).

### 2.3. Study Screening and Selection

In this study, researchers reviewed each other’s search results based on the PRISMA (preferred reporting items for systematic reviews and meta-analysis) research procedure [24] (see Figure 1). In cases of differing opinions, we resolved the issues through consultation. Among the 1072 studies retrieved from individual search engines, we reviewed 555 papers, excluding duplicate papers (N = 517). Finally, we selected 11 papers as study subjects.

### 2.4. Data Extraction

We created a coding sheet to search related research and collect basic information about the papers. We used the coding sheet to record basic information about the papers being subjected to analysis (author, publication year, publication type), experimental information (mean value, standard deviation, number of subjects, leading variables, detailed dependent variables), and special information about the studies (characteristics of experimental subjects, number of times and periods of experiments). Researchers with experience in meta-analysis jointly coded and reviewed the papers according to agreed guidelines. In cases where an individual study required concrete review, we changed the coding through re-examination. By repeating the process, we ultimately agreed on the coded results. PRISMA 2020 for Abstracts Checklist and PRISMA 2020 Checklist can be found in the Appendix A.

### 2.5. Effect Size

Effect size is a common unit used to integrate and analyze individual study results. Researchers often use standardized mean differences, correlation coefficients, and odds ratios as effect sizes. This study used an Excel program to conduct coding and calculated the effect size using Revman (Cochrane Review Manager Software) 5.4 and the R program’s “meta”, “metafor”, and “robis” packages. We calculated the effect size using the standardized mean change difference between the pre- and post-measurements of the experimental and control groups relating to college students’ gratitude promotion programs. The calculation formula for calculating the effect size is [19]:(1)$$d_{diff}=\frac{\underline{D_{trt}}-\underline{D_{con}}}{S_{diff}},\;S_{diff}=\sqrt{\frac{\left(n_{trt}-1\right)S_{trt}^{2}+\left(n_{con}-1\right)S_{con}^{2}}{\left(n_{trt}+n_{con}-2\right)}}$$

According to Cohen’s [25] classification criteria, effect sizes of about 0.2 are small effects, 0.5 are medium, and 0.8 are large. We set the confidence interval at 95%.

### 2.6. Publication Bias

Publication bias is the tendency to preferentially publish statistically significant or highly effective studies [26]. Scholars often use a funnel plot to verify publication bias, and left−right asymmetry in the funnel plot indicates suspected publication bias. We can estimate the studies’ influence by comparing the corrected effect size obtained through the Trim and Fill method [27] after incorporating the missing studies with the current effect size.

### 2.7. Qualitative Evaluation

Qualitative evaluation of studies included in a meta-analysis is a task to increase the credibility and persuasiveness of meta-analyses results [28]. In this study, we conducted a quality evaluation using ROBINS-I [29]. We used the qualitative evaluation form within Revman (Cochrane Review Manager Software) 5.4 to express qualitative evaluations.

## 3. Results

### 3.1. Publication Bias

We investigated publication bias using the R program to check the funnel plot (Figure 2). The results show that four studies did not have bilateral symmetry. Therefore, we verified the influences of the studies through the Trim and Fill method [27].

When we added four more studies (Table 1), the effect size decreased slightly from 0.6671 to 0.4008 while still being statistically significant. However, it did not decrease significantly to affect the study’s validity.

### 3.2. Qualitative Evaluation

Figure 3 shows the risk of graph bias regarding the selected papers, and Figure 4 illustrates the evaluation results of the risk of bias of individual papers. We summarize the risk of bias as follows:The risk of bias due to confounding was low because all studies selected the same subject groups as the participant condition.The risk of bias in participant selection was low because all studies recruited research subjects through one round of recruitment.The risk of bias in classifying interventions was low because all studies had the same intervention time and did not classify interventions based on certain variables.The risk of bias due to deviations from intended interventions was serious because no studies reported the degree of intervention exposure of study subjects.The risk of bias due to missing data was low only in the case of the study conducting in-depth research on those who dropped out during the intervention [29]. We classified the remaining studies as “unclear risk of bias”.The risk of bias in the measurement of outcomes was unclear because all studies measured outcomes using only self-report instruments.The risk of bias in selecting the reported result was low because all papers in this study reported predefined treatment results.

The researchers of this paper have undertaken the task of comprehending and interpreting the studies in consideration of the biases identified above.

### 3.3. Heterogeneity Test

To analyze a valid effect size, we had to first assess homogeneity. Depending on the results, we employed a fixed effect model for homogeneous study groups and a random effect model for heterogeneous ones [28]. We assessed the homogeneity, and the result indicated statistical significance (*p* value < 0.0001), with Q − df > 0, as shown in Table 2, indicating variability among studies.

As for heterogeneity, according to the I-square value, 25% is a small value, 50% is medium, and 75% is large [26,28]. The I-square value in our study was 88.5%, indicating highly heterogeneous study groups. Therefore, our study applied the random effect model.

### 3.4. Study Characteristics

We selected 11 papers as the subjects of meta-analyses, as shown in Table 3, with 683 participants. We found two types of gratitude promotion programs: “gratitude recording” and “gratitude recording and gratitude practice”. The programs’ durations ranged from 1 to 24 weeks.

### 3.5. Overall Effect Size

We measured the overall effect size with a random effects model (Table 4). We found the overall effect size of the gratitude promotion programs for college students to be 0.6671, which is at least a medium effect size.

### 3.6. Effect Sizes of Sub-Areas

Adler and Fagley [40] suggested that various aspects of gratitude manifest in cognition, emotion, and behavior, proposing a gratitude model that is delineated by cognition, emotion, and behavior. Thus, we divided the dependent variables of the college students’ gratitude promotion programs into three areas and sub-elements: cognitive, emotional, and behavioral (Table 5), classifying the dependent variables of the programs, respectively.

The effect sizes by sub-area of the college students’ gratitude promotion programs are in Table 6. The effect size of the cognitive area was the highest at 1.37, while the emotional area was 0.60 and the behavioral area was 0.77.

#### 3.6.1. Effect Sizes of Cognitive Areas

The effect sizes of the cognitive area of the college students’ gratitude promotion programs are in Table 7. The effect size of the gratitude disposition was 1.36, and gratitude recognition was 1.40, all large effect sizes.

#### 3.6.2. Effect Size of Emotional Area

The effect sizes of the emotional area in college students’ gratitude promotion programs are in Table 8. The effect sizes of the emotional area were 0.50 as the overall effect size for positive emotion promotion and 0.90 as the overall effect size for negative emotion reduction. These results indicate that college students’ gratitude promotion programs are more effective in reducing negative emotions than promoting positive ones. In particular, the effect size for happiness promotion among the sub-elements was 0.50, while the effect size for depression reduction was 1.12, indicating that college students’ gratitude promotion programs were more effective in reducing depression than promoting happiness.

The effect size for expectation among the sub-elements of positive emotion promotion was 0.78, larger than the effect size for happiness promotion (d = 0.50) or satisfaction promotion (d = 0.45). The effect size for depression reduction among the sub-elements of reduction in negative emotions was 1.12, a large effect size. However, the effect size for anger reduction was 0.19, a small effect size.

#### 3.6.3. Effect Size of Behavioral Area

The effect sizes of the behavioral area in college students’ gratitude promotion programs are in Table 9. Among the effect sizes of the behavioral area in college students’ gratitude promotion programs, the overall effect size for positive behavior promotion was 0.80, which was larger than that for negative behavior reduction at 0.65. Among the positive behavior promotion’s effect size areas, the effect size for the improving area was 0.93, which was larger than the effect size for building relationships, which was 0.61.

Among the effect size areas for negative behavior reduction, the effect size for blaming was 0.66, while that for getting angry was 0.63, indicating that both showed similar decreases.

### 3.7. Other Effect Sizes

#### 3.7.1. Effect Sizes According to Program Types

The effect sizes of college students’ gratitude promotion programs according to program types are in Table 10. We can categorize college students’ gratitude promotion programs into programs involving simultaneous gratitude recording and practice and those focusing solely on recording gratitude. Among these, the effect size of the programs involving simultaneous gratitude recording and practice was 0.85, slightly larger than that of programs involving only gratitude recording, which was 0.77.

#### 3.7.2. Effect Sizes According to Program Durations

The effect sizes of college students’ gratitude promotion programs according to the program durations are displayed in Table 11. The effect sizes according to the program durations were the smallest at 0.36 in cases, where students followed the programs for less than one week, while the largest were at 1.61 for programs lasting at least 16 weeks.

## 4. Conclusions and Discussion

This study explored the significance and implications of gratitude promotion programs by analyzing their characteristics and effects on college students through meta-analysis. A discussion of the study results obtained by analyzing the 11 studies subject to meta-analysis follows below.

First, we used a random effect model to determine the effect size in the meta-analysis, resulting in an overall effect size of 0.6671 for college students’ gratitude promotion programs. According to Cohen’s [25] effect size interpretation criteria, the result is a medium or large effect size. As Emmons and McCullough’s [41] study shows, the experience of gratitude gave college students a high level of well-being, passion, and concentration. Thus, gratitude promotion programs for college students effectively encourage change and growth in those who worry due to various psychological difficulties and career issues. These results suggest significant implications, indicating that gratitude promotion programs can help college students experiencing psychological and career challenges develop a sense of well-being, passion, and concentration, fostering a positive attitude.

Second, we examined variables frequently investigated in gratitude promotion programs, categorizing them into three sub-areas: cognitive, emotional, and behavioral, as classified by Adler and Fagley [40]. The analysis revealed that the overall effect size of the cognitive area in college students’ gratitude promotion programs was 1.37, the largest among all sub-areas. Weiner [42] defined gratitude as recognizing positive results and the causes of those results—the gratitude promotion programs were very effective in increasing the recognition of gratitude. The results show that rethinking and recognizing gratitude through a gratitude promotion program shows greater effects than other emotions and behaviors. Although everyone is aware of gratitude, concrete, specific activities and expressions of gratitude are relatively weak, and students re-recognized gratitude through a gratitude promotion program. Therefore, all college students should experience gratitude promotion programs.

Third, the effect size of college students’ gratitude promotion programs was above medium in the emotional area (d = 0.60), and the effect size for reducing negative emotions (d = 0.90) was larger than for increasing positive emotions (d = 0.50). In particular, the effect size for depression reduction (d = 1.12) was at least twice that for happiness promotion (d = 0.50), indicating that the gratitude promotion programs were particularly effective in reducing depression. According to a study conducted by McCullough et al. [7], people with a high gratitude disposition have high positive emotions and low depression. This study used concrete values of effect sizes to identify how much the gratitude promotion programs were statistically effective in promoting happiness and reducing depression.

Fourth, in the behavioral area, the effect size for positive behavior promotion (d = 0.80) was larger than that for negative behavior reduction (d = 0.65). In particular, the effect size of improving positive behaviors (d = 0.93) was the highest. Sansone and Sansone [43] stated that gratitude promotes positive behavior and well-being. As with their study, gratitude promotion programs for college students greatly improved their life elements and developed their lives. The results indicate that the gratitude promotion programs are more effective in promoting positive behaviors, such as improving and building relationships, than in reducing negative behaviors, such as blaming or getting angry. Therefore, since gratitude promotion programs significantly foster positive behaviors in college students, colleges should refer to these programs when composing new ones.

Fifth, we analyzed the effect sizes by type of gratitude promotion program. We found a large effect size for programs that simultaneously include gratitude recording and gratitude practice (d = 0.85). This result was larger than the effect size of programs focused solely on gratitude recording (d = 0.77). The finding indicates that gratitude promotion programs in which participants practiced saying they were grateful to surrounding people and writing letters were more effective. This result is consistent with the study, indicating that one can express gratitude well only when combining the positive aspects of the present moment and the experience of being grateful [44]. The results suggest that, when composing a gratitude promotion program for college students, it is necessary to enable them to carry out gratitude recording and gratitude practices simultaneously.

Sixth, we analyzed the effect sizes based on the durations of program performance. The findings revealed that the program groups engaged for at least 16 weeks demonstrated the largest effect size (d = 1.61), whereas the group participating for less than one week showed the smallest effect size (d = 0.36). However, between Weeks 8 and 16, the observed effect size did not increase, indicating no significant increase with longer duration. However, after Week 16, the effect size increases noticeably, indicating the effectiveness of the long-term programs. Therefore, colleges should keep this in mind when creating new programs.

This study has limitations regarding the generalizability of its findings. Specifically, due to the limited availability of gratitude promotion programs for college students in South Korea, the study sample comprised only 11 papers, potentially restricting the broader applicability of the results. Therefore, researchers should conduct more studies on college students’ gratitude promotion programs to supplement this study’s findings. Furthermore, by analyzing gratitude promotion programs across countries and comparing their effects, we can incorporate cultural nuances, thereby obtaining broader and more objective insights. Lastly, we conducted qualitative studies on gratitude in parallel with previous research to add subjective insights into individuals’ experiences with gratitude. [44]. These studies explore the personal significance of gratitude in people’s lives, examining the results alongside quantitative data to enrich the overall understanding of the topic.

This study is the first meta-analysis of the effectiveness of gratitude promotion programs for college students. Its significance lies in its comprehensive analysis of pertinent areas of interest for implementing and applying such programs, including effective strategies, intervention methods, and areas warranting attention. The derived implications offer valuable insights for strategically planning gratitude promotion interventions aimed at enhancing gratitude awareness, mitigating negative emotions, alleviating depression, and fostering positive behaviors among college students.

## Figures and Tables

**Figure 1 behavsci-14-00240-f001:**
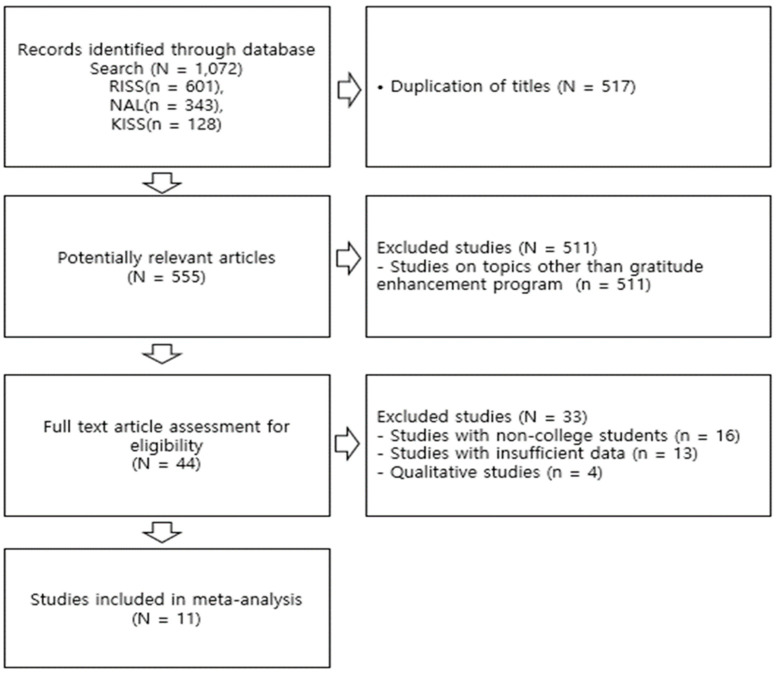
PRISMA flow diagram.

**Figure 2 behavsci-14-00240-f002:**
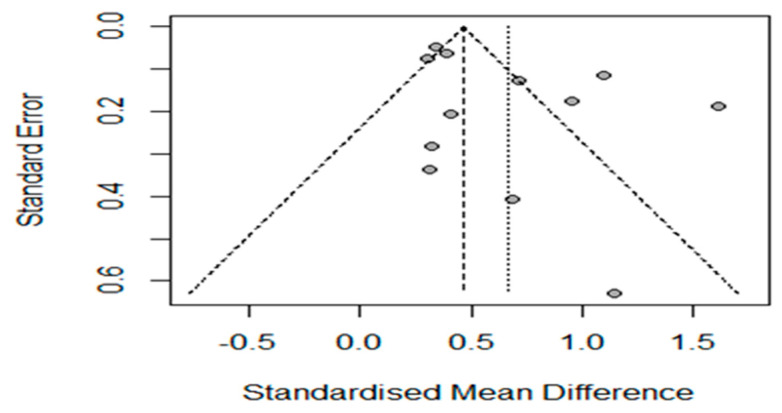
Funnel plot of standard error by the standard difference in means.

**Figure 3 behavsci-14-00240-f003:**
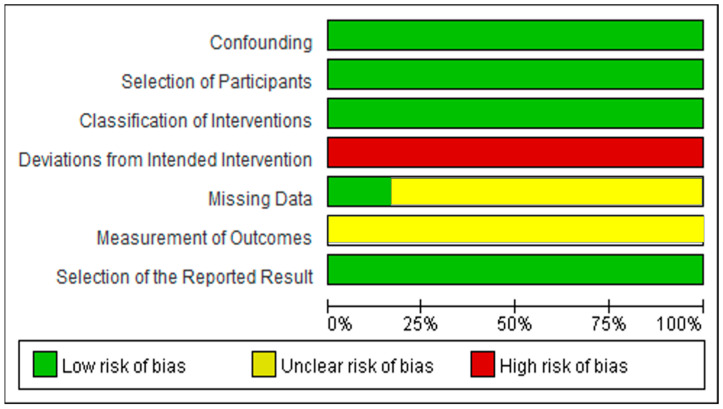
Graph of quality assessment of selected papers.

**Figure 4 behavsci-14-00240-f004:**
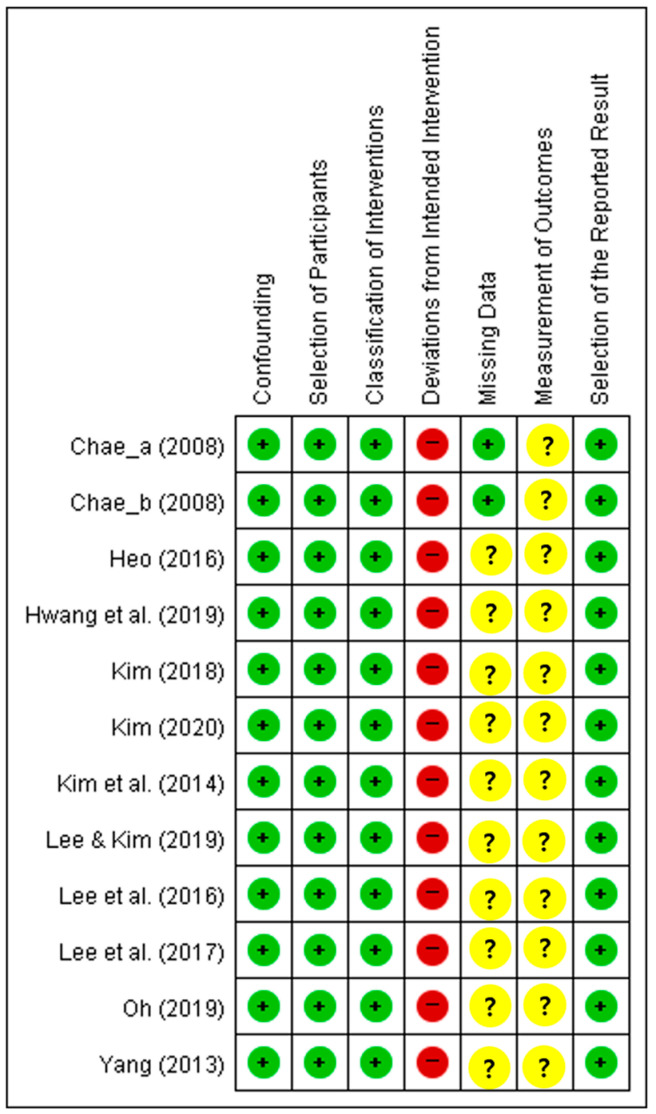
Results of the quality assessment of the selected papers [29,30,31,32,33,34,35,36,37,38,39].

**Table 1 behavsci-14-00240-t001:** Trim and fill method.

	Trimmed Studies	Effect Size	95% CI (Random Effect Model)	
			Lower	Upper
Observed		0.6671	0.4154	0.9188
Adjusted	4	0.4008	0.0841	0.7175

Note: CI = Confidence interval.

**Table 2 behavsci-14-00240-t002:** Homogeneity test.

Model	Q	df	P	I^2^
Fixed	95.57	11	<0.0001	88.5%

Note: Q = observed variance of effect sizes; df = degree of freedom; P = Probability; I^2^ = heterogeneity (ratio of actual variance to total variance).

**Table 3 behavsci-14-00240-t003:** Characteristics of included studies.

No	Researcher	Effect Size	Program Type	Participant Type (School or Major/Gender/Grade)	Total # of Participants (# of Experimental Group Participants)	Total Program Period (Weeks)
1	Chae (2008a) [29]	0.31	Gratitude Recording and Gratitude Practice	A university in S City, South Korea/Male and female students/1st to 4th grades	48 (27)	12
	Chae (2008b) [29]	0.32	Gratitude Recording		44 (23)	12
2	Yang (2013) [30]	1.61	Gratitude Recording and Gratitude Practice	Department of Early Childhood Education/Not mentioned/3rd, 4th grade	89 (42)	24
3	Kim et al. (2014) [31]	0.34	Gratitude Recording and Gratitude Practice	Korea Armed Forces Nursing Academy/Female student/1st grade	77 (40)	4
4	Lee et al. (2016) [32]	0.95	Gratitude Recording	A certain women’s university in South Korea/Female student/Not mentioned	28 (14)	4
5	Heo (2016) [33]	0.71	Gratitude Recording and Gratitude Practice	A certain women’s university in South Korea/Female student/not mentioned	40 (20)	8
6	Lee et al. (2017) [34]	0.41	Gratitude Recording	A university located in Y city, South Korea/Female student/2nd grade	40 (20)	1
7	Kim (2018) [35]	1.10	Gratitude Recording and Gratitude Practice	W University in South Korea/Not mentioned	41 (21)	16
8	Oh (2019) [36]	0.69	Gratitude Recording	College student whose religion is Christianity	50 (25)	11
9	Lee and Kim (2019) [37]	0.38	Gratitude Recording and Gratitude Practice	Department of Early Childhood Education/Female student/1st grade	55 (28)	12
10	Hwang (2019) [38]	0.30	Gratitude Recording	A university in Y City, South Korea/Female student/2nd grade	40 (20)	1
11	Kim (2020) [39]	1.14	Gratitude Recording	A university in G City, South Korea/Not mentioned	131 (68)	14

**Table 4 behavsci-14-00240-t004:** Overall effect size.

	K	ES	95% CI		Q-Value	df	*p*-Value	I^2^
			Lower	Upper				
Total	12	0.6671	0.4154	0.9188	95.57	11	<0.0001	88.5%

Note: K = number of cases of studies; ES = effect size; Q = observed variance of effect sizes; df = degree of freedom; I^2^ = heterogeneity (ratio of actual variance to total variance).

**Table 5 behavsci-14-00240-t005:** Classification according to gratitude areas.

Subareas of the Gratitude Promotion Programs		Sub-Effect
Cognition	Gratitude disposition	Tending to be grateful
	Gratitude recognition	Having a recognition of gratitude
Emotion	Increase in positive emotions	Happiness
		Satisfaction (satisfaction with life, satisfaction with school life, etc.)
		Expectations (expectations of life satisfaction, optimism)
	Decrease in negative emotions	Depression
		Anger
Behavior	Increase in positive behavior	Improving everyday behavior (being self-assertive, evaluating positively, finding meaning, taking care of one’s health)
		Building and improving human relationships (relationships with others, relationships with the opposite sex, family relationships)
	Decrease in negative behavior	Blaming (blaming oneself, blaming others)
		Getting angry

Note: We divided sub-effects into sub-effect groups with related dependent variables indicated in parentheses.

**Table 6 behavsci-14-00240-t006:** Effect sizes of sub-areas.

Division	K	ES	95% CI	
			Lower	Upper
Cognitive area	10	1.37	0.73	2.01
Emotional area	29	0.60	0.48	0.73
Behavioral area	33	0.77	0.60	0.95

Note: K = number of cases of effect sizes; ES = effect size; CI = Confidence interval.

**Table 7 behavsci-14-00240-t007:** Effect sizes of the sub-area cognitive area.

Division	K	ES	95% CI	
			Lower	Upper
Overall effect size of the cognitive area	10	1.37	0.73	2.01
- Gratitude disposition	7	1.36	0.60	2.12
- Gratitude recognition	3	1.40	−0.16	2.95

Note: K = number of cases of effect sizes; ES = effect size; CI = Confidence interval.

**Table 8 behavsci-14-00240-t008:** Effect sizes of emotional area among sub-areas.

Division	K	ES	95% CI	
			Lower	Upper
Overall effect size of the emotional area	29	0.60	0.48	0.73
Overall effect size for positive emotion promotion	21	0.50	0.38	0.62
- Happiness	10	0.50	0.36	0.64
- Satisfaction	9	0.45	0.20	0.69
- Expectation	2	0.78	0.49	1.07
Overall effect size for negative emotion reduction	8	0.90	0.56	1.25
- Depression reduction	6	1.12	0.83	1.42
- Anger reduction	2	0.19	−0.18	0.57

Note: K = number of cases of effect sizes; ES = effect size; CI = Confidence interval.

**Table 9 behavsci-14-00240-t009:** Effect size of behavioral area among sub-areas.

Division	K	ES	95% CI	
			Lower	Upper
Overall effect size of the behavioral area	33	0.77	0.60	0.95
Overall effect size for positive behavior promotion	26	0.80	0.63	0.98
- Improving	16	0.93	0.71	1.15
- Building relationships	10	0.61	0.33	0.89
Overall effect size for negative behavior reduction	7	0.65	−0.09	1.38
- Blaming	4	0.66	−0.58	1.90
- Getting angry	3	0.63	−0.48	1.74

Note: K = number of cases of effect sizes; ES = effect size; CI = Confidence interval.

**Table 10 behavsci-14-00240-t010:** Effect sizes according to the types of gratitude promotion programs.

Division	K	ES	95% CI	
			Lower	Upper
Simultaneous gratitude recording and gratitude practice	44	0.85	0.67	1.03
Gratitude recording only	29	0.77	0.52	1.02

Note: K = number of cases of effect sizes; ES = effect size; CI = Confidence interval.

**Table 11 behavsci-14-00240-t011:** Effect sizes according to program durations.

Division	K	ES	95% CI	
			Lower	Upper
Within one week(7 Days)	4	0.36	0.18	0.54
1~4 weeks	23	0.73	0.56	0.90
4~8 weeks	9	0.71	0.46	0.96
8~16 weeks	25	0.57	0.34	0.79
At least 16 weeks	11	1.61	1.24	1.97

Note: K = number of cases of effect sizes; ES = effect size; CI = Confidence interval.

## Data Availability

Data from the study are available upon request.

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
