# Peer review of "A Meta-Analysis on the Effectiveness of Gratitude Promotion Programs for South Korean College Students"

_behavsci, 2024, doi:10.3390/bs14030240_

Round 1
Reviewer 1 Report
Comments and Suggestions for Authors
Dear authors,
thank you for your manuscript. I have only minor points, I would like you to address.
In the discussion section you describe that the longest durations of interventions have the highest effect sizes and the shortest durations the lowest effect sizes. This results in the impression of a linear increase in effect size over durations. Nevertheless, this is not the case and it would be important to discuss the characteristics of the studies showing these high effect sizes. Is this really an effect of duration or are there other factors resulting in the effect sizes, because the effect sizes are getting smaller over the durations from 1-4, 4-8 to 8-16 weeks.
Second, you state that you conducted qualitative studies to broaden the understanding of the effects at work in gratitude programs. I do not understand the relevance of this statement due to the fact that no results are reported or references supplied.
Third, I think it would be helpful to summarise the derived implications at the end of the manuscript.
Fourth, you showed that all studies are in danger of a bias of deviations from the intended interventions. How should we deal with this bias?
And last, there is a small typo in the abstract, where you used a , instead of a . in the emotional effect size.
Best regards

Reviewer 2 Report
Comments and Suggestions for Authors
Thank you for the invitation to review this manuscript. This study conducts a meta-analysis on gratitude interventions among Korean college students and overall provides valuable information. I have some concerns and suggestions that I hope the authors can clarify.
Major issues:
Why does Table 6 show that k is greater than the total number of studies? Are the authors counting different variables from the same study multiple times? If these are inter-dependent outcomes from a single study, shouldn't they be pooled together?
It is peculiar why the authors directly define the control condition in the methods section as individuals not participating in the intervention (waitlist?). If there are different interventions, especially active controls, it would be worthwhile to analyze them.
The section on "Effect sizes according to program durations" should likely be a meta-regression, rather than conducting separate meta-analyses for each group.
Minor issues:
Table 5 seems somewhat redundant since this information is available in other tables.
Can you specify how many days are included within a one-week intervention? It's mainly to ensure they are structured interventions rather than one-shot interventions (which seems more like a laboratory experiment).
Regarding "cognitive, emotional, and behavioral as classified by Watkins et al." - if this classification is so important and has been introduced by others, consider explaining their specific meanings in the introduction section.
